# Effect of intrinsic foot muscles training on foot function and dynamic postural balance: A systematic review and meta-analysis

**Zhen Wei[1], Ziwei Zeng[1], Min Liu[2]\*, Lin Wang[1]\***

**1** School of Kinesiology, Shanghai University of Sport, Shanghai, China, **2** Shanghai Normal University Tianhua College, Shanghai, China

\* liumin5713039@163.com (ML); wanglin@sus.edu.cn (LW)

## Abstract

This systematic review aimed to analyse the effects of intrinsic foot muscle (IFM) training on foot function and dynamic postural balance. Keywords related to IFM training were used to search four databases (PubMed, CINAHL, SPORTDiscus and Web of Science databases.) for relevant studies published between January 2011 and February 2021. The methodological quality of the intervention studies was assessed independently by two reviewers by using the modified Downs and Black quality index. Publication bias was also assessed on the basis of funnel plots. This study was registered in PROSPERO (CRD42021232984). Sixteen studies met the inclusion criteria (10 with high quality and 6 with moderate quality). Numerous biomechanical variables were evaluated after IFM training intervention. These variables included IFM characteristics, medial longitudinal arch morphology and dynamic postural balance. This systematic review demonstrated that IFM training can exert positive biomechanical effects on the medial longitudinal arch, improve dynamic postural balance and act as an important training method for sports enthusiasts. Future studies should optimise standardised IFM training methods in accordance with the demands of different sports.

## 1 Introduction

Whilst running, the feet act as the starting body part of the lower limb kinetic chain. Aside from functioning as shock absorbers, weight support structures and locomotive effectors [1, 2], the feet can resist deformation, provide a stable base of support and lever the arms to propel the body efficiently [3]. Given that the feet are the most distal aspect of the lower limb and the first part touching the ground [4], many studies have explored their potential mechanism in transmitting ground reaction force and established that impact forces can be distributed through the active modulation of the activity of muscles, such as the plantar flexor, tibialis anterior and calf muscles [3, 5–7].

The main Intrinsic foot muscles (IFMs) are abductor hallucis (ABH), flexor digitorum brevis (FDB) and quadratus plantae (QP). Their principal function is to provide foot stability and

**Funding:** We would like to express our gratitude to the National Natural Science Foundation of China [11572202], which helped us during data collection and analysis.

**Competing interests:** The authors have declared that no competing interests exist.

flexibility for shock absorption [8]; improve dynamic alignment; stiffen the foot arches and stimulate proprioceptors on the sole of the feet [9–12]. IFMs are also categorised as active sub-systems in the foot core system and play an important role in static posture and dynamic activities [1, 13]. During the early stance phase of rearfoot strike running, IFMs are passively stretched as the rearfoot initially touches the ground, and the arch of the foot is slowly compressed to absorb impact energy, which is stored in the relevant plantar elastic structure [6, 12, 14]. In the terminal stance phase, the compressed arch begins to rebound, releasing previously stored elastic energy and providing improved propulsion to runners in the push-off phase [12]. This spring-like mechanism of foot muscles provides 8%–17% of the mechanical energy to the body during every step [2, 15, 16].

IFMs can be trained by using several methods, such as short foot exercise (SFE), toe-posture exercises, towel curl exercises and metatarsophalangeal joint (MPJ) muscle training [17–21]. Amongst these methods, SFE is the most studied because it utilises the IFMs to draw the metatarsal heads back towards the heel whilst minimising distal interphalangeal flexion [18, 22, 23]. Through IFM training, weakened or inhibited IFMs are activated and foot–ankle neuromuscular control is improved [24], which may help prevent running-related injuries, such as plantar fasciitis [25], foot pronation [26], hallux valgus [27] and chronic ankle instability [28].

While a number of isolated studies have shown benefits of IFM, the applicability of these findings is still limited, to date, no previous study has systematically studied these effects nor has a meta-analysis been applied to get an overall estimate of the effect of IFM training. Therefore, the current study aims to identify and determine the effect of IFM training on foot function and dynamic postural balance.

## 2 Methods

### 2.1 Search strategy

This systematic review was conducted in accordance with the PRISMA guidelines [29] and registered in PROSPERO (CRD42021232984). PubMed, CINAHL, SPORTDiscus and Web of Science bibliographic databases were searched by 2 independent authors to identify potentially relevant articles from January 2011 to February 2021. The following search terms were applied in the database search: ('foot muscle' OR 'intrinsic foot muscle' OR 'plantar muscle' OR 'intrinsic flexor foot' OR 'toe muscle' OR 'hallux muscle') AND ('training' OR 'exercise' OR 'strength' OR 'strengthening') AND ('foot function' OR 'foot morphology' OR 'foot structure' OR 'foot posture') AND ('dynamic postural balance' OR 'dynamic balance' OR 'posture stability' OR 'posture control' OR 'postural' OR 'balance'). The Scottish Intercollegiate Guidelines Network criteria were used to describe the include studies [30]. An example of the search strategy for the PubMed database is attached in the supporting information. The search strategy was limited to publications in English.

### 2.2 Study selection

After duplicate articles were removed, the search results were screened independently by 2 authors based on titles, abstracts and full texts on the basis of the following criteria:1) research specific to IFM training as an intervention (treatments, such as SFE, that emphasise the neuromuscular recruitment of the plantar intrinsic foot muscles), 2) having at least 1 desired foot biomechanical parameters (such as navicular drop, foot posture index) and 3) randomised controlled trials (RCTs) or pre-/postintervention studies assessing the effectiveness of an intervention.

## 2.3 Data extraction and analyses

The following data were extracted: (i) author (year), (ii) study design, (iii) population characteristics (e.g. sample size), (iv) interventions (e.g. exercise prescription [sets/repetitions]), (v) outcome characteristics (e.g. foot posture index to describe the parameters of foot function) and (vi) main findings. When the information was unclear, the corresponding author of the study was contacted via email for clarification.

## 2.4 Quality assessment

The methodological quality of the included intervention studies were evaluated by two researchers independently using the Physiotherapy Evidence Database (PEDro) scale [31], which is found to be a reliable and valid measure to evaluate the quality of intervention trials [32], with higher scores indicating lower risk of bias. Each item scoring "yes" contributes 1 point to the total score, except for the first item, which relates to external validity. The total PEDro score thus ranges from 0 to 10 points. Studies with a total score of at least 6 points are considered to be of adequate quality [32, 33]. Notably, if the trials/studies were listed in the PEDro database (https://www.pedro.org.au/), those scores were used in this review.

## 2.5 Quantitative data synthesis and analysis

The training effects were calculated and illustrated based on difference between the pre-intervention and post- intervention parameters using forest plots with Review Manager version 5.3. Random effects models were used to calculate standardised mean differences and 95% confidence intervals (CIs) for the control and experimental groups. The $I^2$ statistic was used to verify heterogeneity ($\chi^2$) between the included studies. The risk of publication bias was also assessed by using funnel plots.

# 3 Results

The electronic database search yielded 249 articles. After duplicates were removed (29 excluded), a total of 220 eligible articles were included. Then, 203 articles were excluded after reviewing the titles and abstracts, reducing the number of articles to 17. After full text screening, 1 article was excluded [34]. Finally, the remaining 16 articles met all the inclusion criteria and were included in this systematic review (Fig 1).

## 3.1 Methodological quality

The results of the risk of bias assessment using the PEDro scale can be found in Table 1. The total scores for the methodological quality ranged from 1 to 8 points. Eight studies [11, 16, 18, 22–23, 26, 35, 36] were moderate quality (PEDro score ≥5) and the others [10, 17, 19, 37–41] were poor quality (PEDro score < 5). The following items were most commonly reported in the articles: random allocation (69%), concealed allocation groups (19%), similar at baseline (63%), blinding of the therapist/subject reported in none of the articles, and blinding of the assessor in ten articles (63%), follow-up > 85% (38%), intention-to-treat analysis (25%), between-group comparison (88%), Point measures and measures of variability (100%).

## 3.2 Study characteristics

The studies included 14 RCTs with sample sizes ranging from 14 to 118 [11, 16–19, 22, 23, 26, 37, 39–41] and 2 pre-/post-test designs with sample sizes of n = 12 and n = 21 [10, 36]. The intervention time of the 16 studies varied: 4 weeks [10, 16, 26], 5 weeks [39], 6 weeks [17, 18,

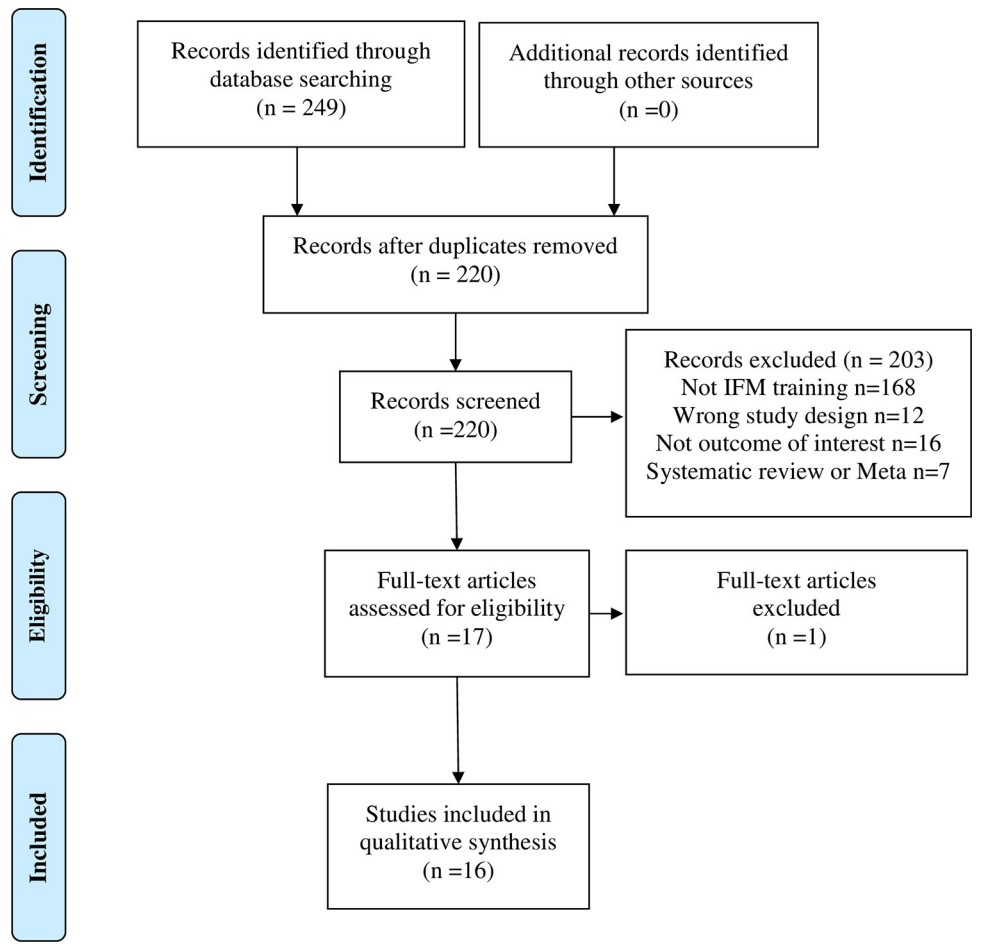

**Fig 1. Flow diagram of literature search.**

35, 41], 7 weeks [19], 8 weeks [11, 22, 36, 38], 9 weeks [40], 10 weeks [37] and 16 weeks [23]. The details of the study and participant characteristics are presented in Table 2.

**3.2.1 Sample characteristics.** The 16 studies included a total of 627 participants. Seven studies included elite long-distance runners (n = 348) [17, 18, 22, 23, 37–39]. Three studies explored the IFMs in patients with pes planus (n = 75) [11, 39, 41], and 6 other studies included only healthy or asymptomatic subjects (n = 207) [10, 16, 19, 26, 36, 40]. The sample sizes of the included studies ranged from 12 to 118 (mean = 39). The validity and statistical conclusions of the study by Hashimoto and Sakuraba (n = 12) were the lowest [38]. Overall, the proportion of males was slightly higher than that of females (54.10%). The age of the runners ranged on average from 20 years old to 45 years old (mean = 27.78), and the body weight of the runners ranged from 50 kg to 76 kg (mean = 67.17). For the foot morphology characteristics, the foot posture indexes in the existing studies ranged from 1 to 10 (mean = 6). Table 3 shows the details of the sample population characteristics.

**3.2.2 IFM foot interventions.** All included studies have various differences in IFM foot interventions. The interventions varied in terms of training methods, exercise prescriptions and timeframes. In broad terms of the training approaches, the included interventions can be categorised as 1. SFE [10, 11, 26, 39, 41], 2. series of foot–ankle muscle training exercises [22,

**Table 1. Results of the risk of bias assessment using the PEDro scale.**

| Study | Eligibility criteria | Random allocation | Concealed allocation | Groups similar at baseline | Subject blinding | Therapist blinding | Assessor blinding | Follow-up > 85% | Intention-to-treat analysis | Between-group comparison | Point measures and measures of variability | Total score |
|---|---|---|---|---|---|---|---|---|---|---|---|---|
| Day and Hahn, 2019 [37] | Yes | Yes | No | Yes | No | No | No | No | No | Yes | Yes | 4 |
| Fraser and Hertel, 2019 [16] | Yes | Yes | No | Yes | No | No | Yes | No | No | Yes | Yes | 5 |
| Goldmann et al., 2013 [19] | Yes | Yes | No | No | No | No | No | No | No | Yes | Yes | 3 |
| Hashimoto and Sakuraba, 2014 [38] | Yes | No | No | No | No | No | No | No | No | No | Yes | 1 |
| Kim and Kim, 2016 [39] | Yes | Yes | No | Yes | No | No | No | No | No | Yes | Yes | 4 |
| Mulligan and Cook, 2013 [10] | Yes | No | No | No | No | No | Yes | No | No | No | Yes | 2 |
| Okamura et al., 2020 [11] | Yes | Yes | No | Yes | No | No | Yes | Yes | Yes | Yes | Yes | 7 |
| Pabon-Carrasco et al., 2020 [26] | No | Yes | No | Yes | No | No | Yes | Yes | Yes | Yes | Yes | 7 |
| Sánchez-Rodríguez et al., 2020 [40] | No | Yes | No | No | No | No | No | No | No | Yes | Yes | 3 |
| Sulowska et al., 2016 [17] | Yes | No | No | No | No | No | Yes | Yes | No | Yes | Yes | 4 |
| Sulowska-Daszyk et al., 2020 [18] | No | Yes | No | Yes | No | No | Yes | No | No | Yes | Yes | 5 |
| Sulowska et al., 2019 [35] | No | No | No | Yes | No | No | No | Yes | Yes | Yes | Yes | 5 |
| Taddei et al., 2020 [23] | Yes | Yes | Yes | Yes | No | No | Yes | Yes | Yes | Yes | Yes | 8 |
| Taddei et al., 2020 [22] | Yes | Yes | Yes | Yes | No | No | Yes | No | No | Yes | Yes | 6 |
| Taddei et al., 2018[36] | Yes | Yes | Yes | No | No | No | Yes | Yes | No | Yes | Yes | 6 |
| Unver et al., 2019 [41] | Yes | No | No | Yes | No | No | Yes | No | No | Yes | Yes | 4 |

Note: Scoring of eligibility criteria specified does not contribute to total score

**Table 2. Characteristics of the included studies.**

| Author (year) Study design | Sample size | Intervention group (IG) | Control group (CG) | Outcome measures | Results |
|---|---|---|---|---|---|
| **Day and Hahn (2019)** [37] RCT | n = 20 competitive distance runners | n = 11 IFM strengthening exercises 4 weeks, 3 times/day, 3 days/week isometric, concentric and eccentric exercises | n = 9 Not prescribed any additional strengthening protocol | • Toe-flexor strength, • MPJ and ankle mechanics • Running economy | • Toe-flexor strength increased • MPJ and ankle mechanics did not change • Running economy did not change |
| **Fraser and Hertel (2019)** [16] RCT | n = 23 healthy recreationally active young adults | n = 11 SFE, toe-spread-out, hallux-extension and lesser-toe-extension 4 weeks, daily, 3 times/day 104 repetitions/day in 12 sets Progression: from sitting to double-limb stance to single-limb stance | n = 12 Received no intervention | • Clinician-assessed motor performance • Participant-perceived difficulty • Ultrasound imaging motor activation measures of the ABH, FDB, quadratus plantae and FHB were assessed during toe-spread-out, hallux-extension and lesser-toe-extension exercises | • Improved motor performance and decreased perceived difficulty when performing the exercises • No changes in the ultrasound imaging measures of IFM activation in the IG compared with those in the CG |
| **Goldmann et al. (2013)** [19] RCT | n = 27 healthy males | n = 15 Heavy resistance toe flexor muscle training with 90% of the maximal voluntary isometric contraction (MVIC) 7 weeks, 4 times/week 4 sets of 5 repetitions (3 s contraction, 3 s relaxation) | n = 12 No training programme and continued their daily activities | • Maximal MPJ and ankle plantar flexion moments during MVICs were measured • Motion analyses were performed during barefoot walking, running and vertical and horizontal jumping | • MPJ plantar flexion moments in the dynamometer, external MPJ dorsiflexion moments and jump distance in horizontal jumping increased significantly |
| **Hashimoto and Sakuraba (2014)** [38] Pre/post | n = 12 healthy males | Toe flexion interphalangeal/MPJ 3 kg load 8 weeks, 3 days/week 200 repetitions/day | - | • Digital grip dynamometer • Foot arch measurements (longitudinal and horizontal planes) during static standing using the Berkemann footprint • Dynamic test items: single-leg long jump, vertical jump and 50 m dash | • Significant changes observed for intrinsic foot flexor strength scores, foot arches, vertical jumping, single-leg long jumping and 50 m dash time |
| **Kim and Kim (2016)** [39] RCT | n = 14 university students with flexible flatfoot | n = 7 SFE 5 weeks, 3 times/week, 30 min each time | n = 7 Arch support insoles | • ND tests • Y-balance tests | • SFE group showed significant decreases in ND tests • SFE group and arch support insole group showed significant increases in Y-balance tests |
| **Mulligan and Cook (2013)** [10] Pre/post | n = 21 asymptomatic subjects | SFE: 5 s hold up to 3 min per day for approximately 4 weeks, daily, 30 repetitions/day Progression: from sitting to double-limb stance to single-limb stance until reaching 3 min | - | • ND difference between the seated and standing navicular positions • AHI calculated by dividing the dorsum foot height by the truncated length of the foot in seated and standing positions to obtain a ratio • Intrinsic foot musculature test • SEBT | • Subject ND decreased by a mean of 1.8 mm at 4 weeks and by 2.2 mm at 8 weeks • AHI increased from 28% to 29% • Grade of IFM performance during a static unilateral balancing activity improved from fair to good • Significant improvement during a functional balance and reach task in all directions with the exception of an anterior reach |

(*Continued*)

**Table 2.** (Continued)

| Author (year) Study design | Sample size | Intervention group (IG) | Control group (CG) | Outcome measures | Results |
|---|---|---|---|---|---|
| **Okamura et al. (2020)** [11] RCT | n = 20 patients with pes planus | n = 10 8 weeks SFE 3 times/week 3 sets/time 10 repetitions/sets each repetition was held for 5 s with a 45-s rest period between sets Progression: from sitting to double-limb stance to single-limb stance | n = 10 Received no intervention | • Foot kinematics during gait, including dynamic ND—the difference between navicular height at heel strike and the minimum value • Time at which navicular height reached its minimum value • Three-dimensional motion analysis to assess static foot alignment via the FPI and ND test • Thickness of the intrinsic and extrinsic foot muscles was measured by using ultrasound | • FPI scores with regard to calcaneal inversion/eversion improved significantly • Time required for navicular height to reach the minimum value decreased significantly |
| **Pabon-Carrasco et al. (2020)** [26] RCT | n = 85 asymptomatic participants | n = 42 SFE reinforcement: maintain the position of maximum shortening for 30 s from sitting to standing position to standing unipodal 4 weeks, daily, 50 repetitions/day | n = 43 Nonbiomechanical function exercise | • Foot posture was evaluated twice via the ND test • FPI | • Comparison of foot posture before and after training found no statistically significant differences between the experimental group and CG • FPI was modified in both groups with respect to its initial state and the ND value decreased |
| **Sánchez-Rodríguez et al. (2020)** [40] RCT | n = 36 healthy adults | n = 18 9-week intrinsic and extrinsic foot and core muscle strength program 2 sessions/week 40 min/session | n = 18 Received no intervention | • FPI scores | • IG showed significantly reduced FPI by 1.66 points, whereas the score of the CG was the same as that preintervention |
| **Sulowska et al. (2016)** [17] RCT | n = 25 long-distance runners | n = 12 SFE and balanced loading of the 3 support points of the foot 6 weeks, daily, 2 times/day, 15 min/time Progression: from sitting to standing to half-squat | n = 13 Vele's forward lean and reverse tandem gait exercise | • FPI scores • FMS tests | • Significant improvement in the FPI-6 (inversion/eversion of the calcaneus after SFE intervention) |
| **Sulowska-Daszyk et al. (2020)** [18] RCT | n = 80 long-distance runners | n = 48 SFE, balanced loading of the 3 support points of the foot 6 weeks, daily, 30 min, repeated 30 times Progression: increasing the load and level of difficulty every 2 weeks in seated, standing and half-squat positions | n = 32 Received no intervention | • Quality of movement patterns with the FMS was evaluated before and after intervention • Muscle flexibility was evaluated before and after intervention | • Significantly increased FMS values in individual tasks and in the total score after 6 weeks • Significant improvement in muscle flexibility at baseline and after 6 weeks (e.g. external rotation muscles) |
| **Sulowska et al. (2019)** [35] RCT | n = 47 long-distance runners | n = 27 with neutral foot 6 weeks, daily basis for 30 min Vele's forward lean and reverse tandem gait exercise, SFE and stability exercise | n = 20 with slight and increased pronation | • Knee flexor and extensor torque, work • Power on isokinetic dynamometer • Running-based anaerobic sprint test | • Increased values of the peak torque of knee flexors • Increased values of maximum power |

(*Continued*)

**Table 2.** (Continued)

| Author (year) Study design | Sample size | Intervention group (IG) | Control group (CG) | Outcome measures | Results |
|---|---|---|---|---|---|
| **Taddei et al. (2020)** [23] RCT | n = 118 healthy runners | n = 57 16 weeks of foot core training (8-week training course, followed by 8 weeks of remotely supervised training) 4 times/week (1 time by a physical therapist and 3 times given through online videos) Both groups were instructed to perform their respective exercises 3 times/week up to the end of the 12-month follow-up | n = 61 5 min placebo static stretching protocol 3 times/week on the basis of online descriptions | • Assessments consisted of 3 separate biomechanical evaluations of foot strength • FPI • Weekly report on each participant's running distance, pace and injury incidence over 12 months | • CG participants were 2.42 times more likely to experience an RRI within the 12-month study period • Time to injury was significantly correlated with FPI and foot strength gain scores • Foot exercise program showed evidence of effective RRI risk reduction in recreational runners at 4–8 months of training |
| **Taddei et al. (2020)** [22] RCT | n = 28 healthy recreational long-distance runners | n = 14 8-week foot–ankle exercise during weight-bearing activities (with a physiotherapist once a week and at least 3 times at home over the entire course of the study) | n = 14 8 weeks of 5 min warm-up and full body muscle stretching protocol | • Hallux and toe strength • Foot function • Cross-sectional area and volume of the ABH, ADM, FDB and FHB • MLA range of motion and stiffness • Vertical and anteroposterior propulsive impulses during running | • Volume of all investigated muscles and muscles for vertical propulsive impulse during running increased in the IG relative to those in the CG Correlations were found between vertical propulsive impulse and volume of ABH, ADM and FDB |
| **Taddei et al. (2018)** [36] RCT | n = 30 healthy recreational long-distance runners | n = 15 8-week foot–ankle muscle strength (trained in weekly sessions by a physiotherapist and instructed to perform the same exercises at home at least twice a week) | n = 15 5 min placebo warm-up and muscle stretching protocol | • Hallux and toe muscle strength using a pressure platform • Foot muscle cross-sectional area using magnetic resonance imaging • Foot kinematics during running using 3D gait analysis | • Cross-sectional area of the ABH and FDB increased significantly at 8 weeks in the IG |
| **Unver et al. (2019)** [41] RCT | n = 41 patients with pes planus | n = 21 6-week SFE training daily | n = 20 Received no intervention | • ND • FPI • Foot pain • Disability • Plantar pressures | • ND, FPI, pain and disability scores significantly decreased • Maximum plantar force of the midfoot significantly increased |

Note: SFE, short foot exercise; ABH, abductor hallucis; FDB, flexor digitorum brevis; ADM, abductor digiti minimi; FHB, flexor hallucis brevis; MPJ,

Metatarsophalangeal joint; MLA, Medial longitudinal arch; FPI, foot posture index; ND, navicular drop; SEBT, star excursion balance test; FMS, functional movement

screen; AHI, arch height index

RCT, randomised controlled trial; IG, intervention group; CG, control group.

23, 36, 40], 3. SFE and stability training of the foot [17, 18, 35], 4. SFE and toe/hallux-extension exercises [16, 37] and 5. interphalangeal joint and MPJ loading exercises [19, 38].

## 3.3 Outcome measures

For the characteristics of IFMs before and after training, the included studies measured the muscle activation ratio (contracted measurement/resting measurement) [16] through ultrasonographic imaging and the thickness [11], cross-sectional areas [22, 36] and volume [22] of the foot muscles by using magnetic resonance imaging (Table 2). Additionally, IFM training was measured directly through the hallux or toe muscle strength test [19, 22, 23, 36, 37, 41] and intrinsic foot musculature test [10] by using a custom-made dynamometer.

Medial longitudinal arch morphology was evaluated on the basis of the navicular drop [10, 11, 26, 39, 41] and arch height index [10]. These parameters were verified to provide accurate changes in the medial longitudinal arch. Six studies (n = 326) used the multidimensional and

**Table 3. Sample sizes and participant characteristics for each included study.**

| Included studies | N | | Sex, M/F | | Age, y | | Height, cm | | Body mass, kg | | BMI, kg/m² | | FPI | |
|---|---|---|---|---|---|---|---|---|---|---|---|---|---|---|
| | INT | CON | INT | CON | INT | CON | INT | CON | INT | CON | INT | CON | INT | CON |
| Day and Hahn [37] | 11 | 9 | NR | NR | 24(6) | 30(12) | 173(1) | 172(1.1) | 60(8) | 62(8) | NR | NR | NR | NR |
| Fraser and Hertel [16] | 11 | 12 | 6/5 | 6/6 | 23.6 (6.6) | 19.6 (1.2) | 170.9 (11.5) | 166.5 (13.8) | 70.5 (12.0) | 64.9 (9.5) | 24.0 (2.0) | 23.5 (3.1) | 6.7 (4.2) | 6.0 (3.9) |
| Goldmann et al. [19] | 15 | 12 | 15/0 | 12/0 | 24.0 (4.0) | 26.0 (2.0) | 185.0(7.0) | 181.0(6.0) | 77.0(9.0) | 77.0(5.0) | NR | NR | NR | NR |
| Hashimoto and Sakuraba [38] | 12 | - | 12/0 | - | 29(5) | - | 172.5(7.3) | - | 64.9 (12.8) | - | NR | - | NR | - |
| Kim and Kim [39] | 7 | 7 | 6/1 | 4/3 | 24.0 (1.9) | 24.1 (1.5) | 172.2(6.9) | 167.0(6.7) | 68.2 (12.9) | 63.3 (17.6) | NR | NR | NR | NR |
| Mulligan and Cook [10] | 21 | - | 3/18 | - | 26.1 (3.7) | - | 168.4(7.1) | - | 69.3 (13.6) | - | NR | - | NR | - |
| Okamura et al. [11] | 10 | 10 | 1/9 | 2/8 | 19.7 (0.9) | 20.2 (1.5) | 158.6(6.1) | 159.5(8.8) | 49.7(4.5) | 53.7(7.7) | 19.8 (1.4) | 21.1(2.1) | 9.7(1.9) | 9.0(2.1) |
| Pabon-Carrasco et al. [26] | 42 | 43 | 24/18 | 18/25 | 19.5 (0.4) | 20.9 (1.1) | NR | NR | NR | NR | 24.1 (4.2) | 21.65 (3.4) | 6.8(0.6) | 6.35 (0.3) |
| Sánchez-Rodríguez et al. [40] | 18 | 18 | 7/11 | 8/10 | 23.6 (5.9) | 21.6 (1.9) | NR | NR | NR | NR | 23.2 (3.2) | 23.9(2.6) | 8.1(1.7) | 8.0(1.2) |
| Sulowska et al. [17] | 12 | 13 | NR | NR | NR | NR | NR | NR | NR | NR | NR | NR | NR | NR |
| Sulowska-Daszyk et al. [18] | 48 | 32 | 31/17 | 26/6 | 32.5 (6.8) | 33.4 (7.8) | 175.0(8.7) | 177.7(7.9) | 69.8(9.7) | 71.0 (10.6) | NR | NR | NR | NR |
| Sulowska et al. [35] | 27 | 20 | NR | NR | NR | NR | NR | NR | NR | NR | NR | NR | NR | NR |
| Taddei et al. [23] | 57 | 61 | 28/29 | 33/28 | 40.5 (7.9) | 41.3 (6.8) | 167.4(8.2) | 171.0(9.1) | 68.2 (12.3) | 72.1 (13.2) | 24.2 (2.9) | 24.5(3.2) | 1/0 | 2/2 |
| Taddei et al. [22] | 14 | 14 | 5/9 | 9/5 | 41.9 (7.4) | 41.6 (6.0) | 166.4(7.8) | 169.4(9.2) | 68.3 (12.7) | 75.1 (13.9) | NR | NR | 2.5/1.5 | 2.5/2 |
| Taddei et al. [36] | 16 | 15 | 11/5 | 7/8 | 39.4 (8.5) | 44.8 (8.7) | 169.6 (9.4) | 168.7 (8.8) | 70.7 (12.4) | 67.8 (12.7) | NR | NR | 1/0 | 2/2 |
| Unver et al. [41] | 21 | 20 | 5/16 | 11/9 | 21 (1) | 21.4 (1.7) | NR | NR | NR | NR | 22.9 (3.3) | 23.1(1.9) | 9.0(1.5) | 8.4(2.0) |

Note: NR, not reported; BMI, body mass index; FPI, foot posture index; INT, intervention group; CON, control group

comprehensive evaluation of the foot posture index for the pronation/supination of the feet [11, 17, 23, 26, 40, 41]. This index has been verified to have clinical applications in assessing the risk of injury in athletes (Table 2).

Dynamic postural balance was evaluated by using some function tests, such as the functional movement screen test [17, 18], the star excursion balance test [10, 39] and clinician-assessed motor performance [16] (Table 2). Additionally, Fraser and Hertel [16] explored the participants' perceived difficulty during the toe-spread-out, hallux-extension and lesser-toe-extension tests.

## 3.4 Data analysis

The results of cross-sectional area indicated no significant effect on the muscle characteristics of the Abductor hallucis (ABH) (P = 0.07), Abductor digiti minimi (ADM) (P = 0.08), Flexor digitorum brevis (FDB) (P = 0.22) and Flexor hallucis brevis (FHB) (P = 0.20).

IFM training was observed to have a significant effect on the medial longitudinal arch. The navicular drop (P = 0.02) and foot posture index (P = 0.0003) after IFM intervention had significantly decreased relative to those after the control treatment. The mean difference was −1.97 (95% CI: -3.57–-0.36) for the navicular drop (Fig 2) and -0.69 (95% CI: -1.06–-0.32) for

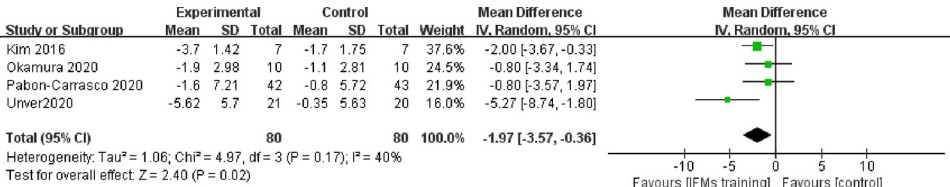

**Fig 2. Forest plot illustrating navicular drop of meta-analysis comparing IFMs training with control group.**

the foot posture index (Fig 3). No significant heterogeneity was observed amongst studies (navicular drop: $I^2 = 40\%$, P = 0.17; FPI: $I^2 = 35\%$, P = 0.18). The bias funnel plots of the navicular drop (Fig 4) and foot posture index (Fig 5) did not suggest evidence of publication bias in the studies included in this meta-analysis.

A significant difference was found for dynamic postural balance after intervention. Although various function tests were included and were difficult to synthesise, the included studies all demonstrated that IFM training can exert positive effects on dynamic postural balance.

## 4 Discussion

This systematic review performed a meta-analysis to summarise the current studies that explored the effect of IFM training on foot biomechanical outcomes. Although potential differences in IFM intervention type, time or frequency may contribute to the potential heterogeneity of the included studies, the current studies verified that IFM training would bring positive biomechanical effects and ameliorate dynamic postural balance.

Four included studies (n = 102) explored the effects of IFM training on muscle morphology [11, 16, 22, 36]. However, no significant difference was found in terms of the parameters of IFM thickness, cross-sectional area and volume. Possible explanations for these discrepancies maybe explained by previous studies indicating that small volumes of IFMs are covered by plantar fascia, which would bring barrier to detect the slight changes in foot muscles [11]. Additionally, Taddei et al. [22, 36] also established proposed that ABH, FDB and FHB have various origins and insertions, different lever arm lengths and may be trained from different degrees during the intervention. Thus, the strength change of single IFMs may be different and hard to detect. In light of the difficulty in measuring the small cross-sectional area of IFMs, future studies should utilise advanced technology, such as magnetic resonance imaging, to measure the IFM fat infiltration and cross-sectional area after training.

Another direct parameter used to describe muscle characteristics is IFM strength. Considering that no gold standard for measuring IFM strength exists [42], the studies included in this review measured IFM strength by applying various approaches, such as pressure platforms [22, 23, 36, 41] or the intrinsic foot musculature test [10]. Although Day and Hahn [37] verified the positive effect of IFM training on muscle strength, no significant difference was found after pooling the data of the included studies. One possible reasons for this conflicting result

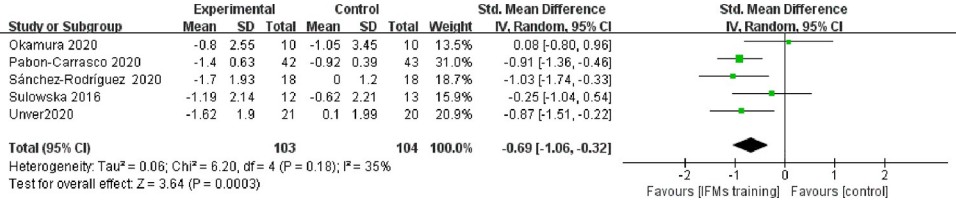

**Fig 3. Forest plot illustrating foot posture index of meta-analysis comparing IFMs training with control group.**

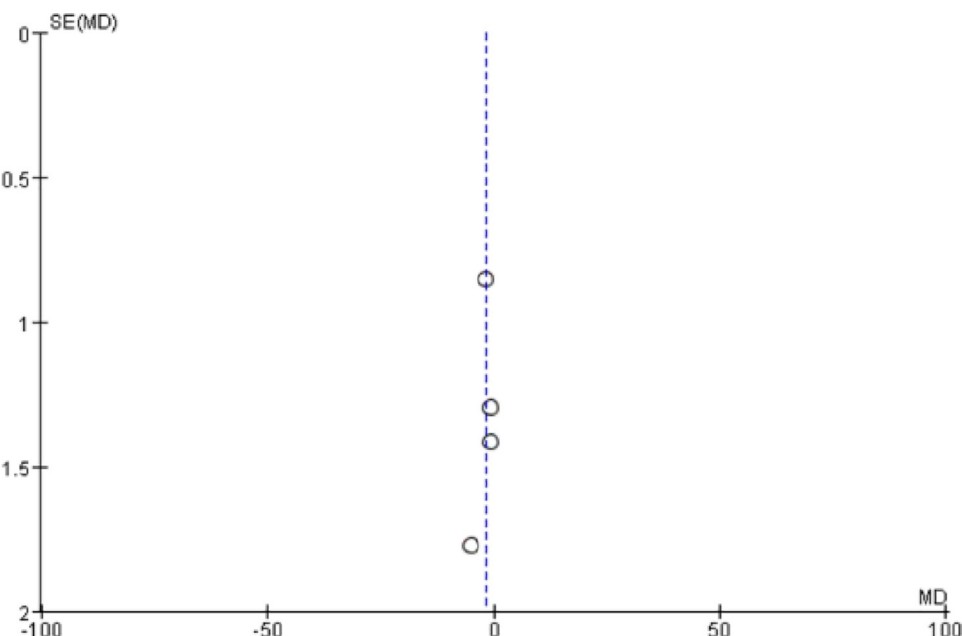

**Fig 4. Funnel plots showing publication bias among studies used to compare IFMs training and control groups.**

might be related to the compensation of extrinsic muscles, such as the tibial posterior muscle [43, 44]. Although studies have attempted to avoid possible interference factors by placing the lower limb in a special position, external muscles are still involved during the test. Unlike previous results reported that enhanced IFM strength can provide additional propulsive impulses, making the foot similar to a stiffened spring during late stance [44, 45], the current study also did not observe any differences. Hence, the actual effect of strength training on IFMs needs to be studied further.

The navicular drop and arch height index are 2 common parameters that describe medial longitudinal arch morphology and dynamic function. IFM exercise is believed to activate weakened IFMs and increase IFM recruitment by intensifying and optimising the tension of the

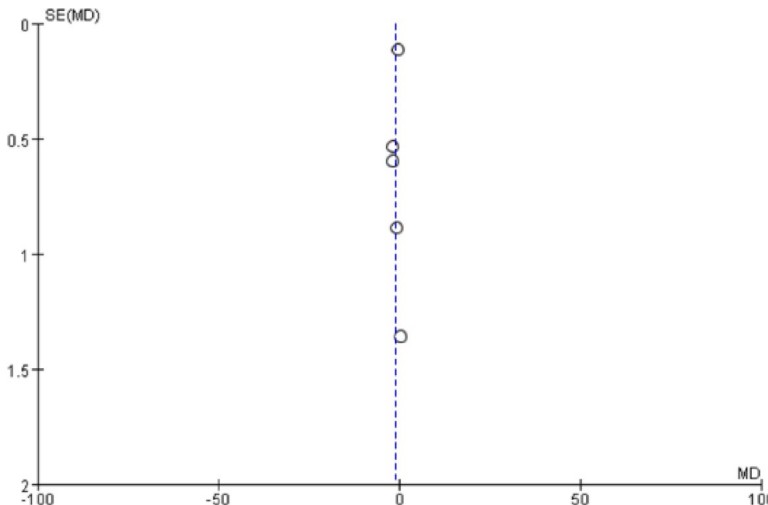

**Fig 5. Funnel plots showing publication bias among studies used to compare IFMs training and control groups.**

medial longitudinal arch, thereby preventing the excessive lowering of the medial longitudinal arch [10, 46] and related running injuries. This systematic review included 4 studies (n = 80) that utilised the navicular drop [10, 11, 26, 41] and 1 study (n = 21) that utilised the arch height index [10] to explore the changes in the medial longitudinal arch. The included studies all demonstrated that the morphology and function of the medial longitudinal arch significantly improved after several weeks of intervention. Even though IFM morphology and single muscle strength showed no significant difference, the overall effect of the medial longitudinal arch was improved. This finding indicated that IFM training can be recommended as an effective measure to improve medial longitudinal arch function and might provide further benefits to people with pes planus. Moreover, the foot posture index is a validated measure for quantifying foot posture. Five studies (n = 103) demonstrated that foot posture index can rectify abnormal lower extremity alignment and stress on the foot and related structures [11, 17, 26, 40, 41].

Amongst the included studies, several measured the dynamic postural balance after IFM training by utilising the functional movement screen [17, 18], star excursion balance test [10], clinician-assessed motor performance and 1-legged long jumping [18]. Although various methods can be applied to assess dynamic postural balance, the results of the included studies established that IFM training has significant positive effects compared with other interventions. Additionally, of the 2 included studies that subjectively assessed IFM training difficulty and foot pain in different situations, the difficulty in motor function perceived by the participants seemed uncomplicated, and the pain in the pes planus was alleviated.

The main limitation of this systematic review is that the included studies varied in terms of their interventions' approaches, time and frequency and their participants' characteristics, this variation might compromise this study. In addition, the included studies utilised different methods for assessing IFM strength and dynamic postural balance. Potential heterogeneity and slight publication bias in the analysis may exist. Therefore, caution is warranted when interpreting the findings of this study.

## 5 Conclusion

Although the interventions of the included studies seemed inconsistent, this systematic review demonstrated that IFM training can exert positive biomechanical effects on the medial longitudinal arch, improve the postural balance of the lower limbs and act as an important training method. Future studies should optimise standardised training methods in accordance with the demands of different sports.

## Supporting information

**S1 Table. Searching terms.**
(DOCX)

**S2 Table. PRISMA 2020 checklist.**
(DOCX)

**S3 Table. International prospective register of systematic reviews (CRD42021232984).**
(PDF)

## Acknowledgments

We would like to express our gratitude to those who helped us during the process of this article. I gratefully acknowledge the help of my supervisor Lin Wang, who have offered me valuable suggestions in this article.

## Author Contributions

**Conceptualization:** Zhen Wei, Lin Wang.

**Data curation:** Zhen Wei, Ziwei Zeng.

**Formal analysis:** Zhen Wei, Ziwei Zeng.

**Funding acquisition:** Lin Wang.

**Investigation:** Zhen Wei.

**Methodology:** Zhen Wei, Ziwei Zeng.

**Supervision:** Min Liu, Lin Wang.

**Writing – original draft:** Zhen Wei.

**Writing – review & editing:** Zhen Wei, Lin Wang.

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
