## [Decision Letter · Decision Letter 0]

21 Sep 2021

PONE-D-21-23366Effect of intrinsic foot muscle training on foot function and dynamic postural balance: a systematic review and meta-analysisPLOS ONE

Dear Dr. Wang,

Thank you for submitting your manuscript to PLOS ONE. After careful consideration, we feel that it has merit but does not fully meet PLOS ONE’s publication criteria as it currently stands. Therefore, we invite you to submit a revised version of the manuscript that addresses the points raised during the review process.The reviewers provided detailled feedback. Please address these points. 

We look forward to receiving your revised manuscript.

Kind regards,

Peter Andreas Federolf

Academic Editor

PLOS ONE

 “This work was supported by the National Natural Science Foundation of China  [11572202]”

4. Thank you for stating the following in the Funding Section of your manuscript:

“This work was supported by the National Natural Science Foundation of China [11572202”

We note that you have provided information within the Funding Section. Please note that funding information should not appear in other areas of your manuscript. We will only publish funding information present in the Funding Statement section of the online submission form.

“This work was supported by the National Natural Science Foundation of China [11572202]””

Reviewers' comments:

Reviewer's Responses to Questions

**Comments to the Author**

1. Is the manuscript technically sound, and do the data support the conclusions?

Reviewer #1: Yes

Reviewer #2: Partly

2. Has the statistical analysis been performed appropriately and rigorously? 

Reviewer #1: Yes

Reviewer #2: I Don't Know

3. Have the authors made all data underlying the findings in their manuscript fully available?

Reviewer #1: Yes

Reviewer #2: Yes

4. Is the manuscript presented in an intelligible fashion and written in standard English?

Reviewer #1: Yes

Reviewer #2: No

5. Review Comments to the Author

Reviewer #1: The authors present a systematic review study on the effectiveness of interventions that aim to improve the functioning of the Intrinsic Foot Muscles. This work has potential relevance in the prevention of foot and ankle injuries and provides a better understanding of the functional anatomy of the foot and the effects of training thereon. The manuscript overall is of sound quality, however it needs some strict improvements in the methods section in order to become eligible for publication.

MAJOR ISSUES

Aims. The aim of the current study is somewhat unclear: “Nonetheless, current studies on IFM training are cross-sectional, and evidence to support this claim is lacking. Therefore, the aim of this systematic review is to identify and determine the effect of IFM training on foot function and dynamic postural balance” (line 48-50).

This needs to be further clarified. If all past studies are cross sectional, what will be the added value of this review? In fact, later you find 13 RCTs that have been done. Why do these not feature in the introduction?

This issue is further confused at the start of the discussion where the aim seems to have been to summarize the current studies. Furthermore, judging from the methods and the results, the aim seems to have been to assess the effect of IFM training on foot biomechanical outcomes using a meta-analysis. Please clarify.

Methods. The major issues of the current manuscript lie in the methods section. Although the authors provide an adequate description of the methods to see that their approach is well thought out, there is insufficient details, and sometimes inconsistencies, that limit the reproducibility of the approach. Relating to this, the following issues will need to be addressed:

1. In the description of the search (line 57-60), it would be good to add some specification of the BOOLEAN operators used in the search.

2. In the supplementary file specifying the search, it should be specified what type of search was performed (e.g. title/abstract/keywords, topic, mesh terms) and it should be specified how the search was adapted to the different search engines.

3. There is some confusion in the use of quotation marks. In the main text, single quotation marks are used, in the supplementary file these are double marks. Please specify what was used in the searches, as these have a different meaning in the search engines.

4. The authors note the inclusion criterium ‘Research specific to IFM training as an intervention’ (line 66). As the authors mention, these muscles have a function many whole-body activities (e.g. walking), so arguably can be trained with many whole body activities as well (e.g. walking, running, whole body vibration training). How was IFM training defined for inclusion in the review?

5. No limitations were introduced to the search dates and yet only articles from after 2013 were included. This might just result from the novelty within the research field, but it could also indicate a fatal flawy in the search strategy. That is, it might be that the field evolved and that now different terms are used compared to 20 years ago. On example might be the following study: DOI: 10.1123/jsr.21.4.327. Please comment on this issue.

6. The authors state ‘When possible, the results of the included studies were pooled for meta-analysis using RevMan 5.3.’ (line 84). How was it determined whether this was possible? It would be helpful to inform the reader on this process. How come only 6 of the 15 papers ended up in the meta-analysis?

7. The prospero registration introduces the usage of the PEDro scale, in the methods section a ‘Checklist for Measuring Quality’ is introduced and in the results the Modified Downs and Black Quality Index is reported. Please resolve these inconsistencies and please specify why the prospero protocol was not adhered to.

MINOR ISSUES

1. The use of abbreviations drastically limits the readability of the manuscript. Following PLOS one author guidelines, the use of abbreviations should be limited, and non-standard abbreviations should not be used unless they appear at least three times in the text. While I believe that most abbreviations are used more than thrice, this is mainly through their use in table 2. I would advise to limit abbreviating terms in the main text and only use abbreviations in the table.

2. Line 22. I would suggest to use the term ‘effector’ rather than ‘organ’

3. Line 27. With the introduction of IFMs, I would advise to name which muscles are meant specifically.

4. Line 39-40. With introduction of the training methods, it would be helpful if the authors could explain in a bit more detail what these training methods are about and in particular where they are different from each other.

5. Line 65-66. “The remaining studies were then reviewed as full texts on the basis of the following criteria: …” - The present wording makes it appear like the title and abstract screening were based on different criteria. Please clarify whether this was the case.

6. Line 91-92. “After full text screening, one article as clinical pearl was excluded” – personally I am unfamiliar with the term ‘clinical pearl’ and why this should lead to exclusion. This might be just me, but in any case a bit further clarification might be helpful.

7. Line 135. Descriptive stats are only provided for runners. How about the non-runner participants?

8. Line 148-149. The information about intervention duration is duplicated after section 3.2 and does not need to be mentioned here.

9. Line 153. Please define extrinsic foot muscles.

10. Line 169. Please define CSA

11. Line 199. Please provide a more extensive/descriptive caption for figure 4.

12. Line 216. The authors mention advance technologies. Can they provide some example of such technologies?

13. Line 238. I believe this should be ‘though’ instead of ‘through’.

Reviewer #2: The goal of this manuscript was to develop a systematic review and a meta-analysis systematic review to identify the effect of intrinsic foot muscles training on foot function and dynamic postural balance.

The manuscript is relatively well grounded, it deals with a relevant topic to be investigated and deserves to be published. In its current state, the level of English throughout your manuscript need careful revision. I suggest a fluent English speaker read and correct grammatical, word choice, and syntax errors throughout the manuscript.

Although paper is straightforward but there are many suggestions/comments that I addressed throughout the manuscript.

There is a critical point in the methodology that is pooling data from studies that used different intervention approaches and time of intervention, and thus, the results are not expected to be the same. Pooling the data into a meta-analysis, it groups studies even if the results are not coming from the same approach of IFM training. Pooling data that came from different interventions, exercises and times of intervention and populations lead to huge heterogeneity that will certainly compromise a meta-analysis. It is pivotal to choose same interventions to be included in a meta-analysis and only then we could have an unbiased statistical analysis. This should be acknowledged and maybe exclude the meta-analysis from the manuscript. it does not add much as the only two variables that could be pooled did not achieve power enough to conclude anything robust.

Language issues examples (there are other weird sentences and word choices that need revision):

1. Please, consider reviewing the English language and grammar by a native reviewer. The writing compromises the understanding of parts of the manuscript.

2. Introduction: “Whilst running, the feet act as the starting body parts…” – correct to body PART.

3. Introduction: “Given that the feet are the most proximal aspect of the lower limb…”- correct to most DISTAL. The feet are distal and not proximal part of LL.

4. Introduction: “ground impact force (GRF)…” – correct to ground REACTION force.

5. Introduction: “During the early stance phase in running…” – correct to OF REARFOOT STRIKE running

6. Introduction: “… and providing better propulsion for runners in the swing phase [12].” – it provides better propulsion in the PUSH OFF phase, not swing phase!

Abstract and Introduction

1. Please, provide the PROSPERO registry number in the abstract.

2. At the end of the second paragraph of the introduction, the authors provide a closing sentence concluding all previous sentences, however it does not make sense. “Therefore, strengthening of IFMs has been suggested for the prevention and treatment of foot and ankle sports injuries [16].” The previous sentences describe all important roles of the foot and its intrinsic muscles, but they do not provide any evidence that strengthening the intrinsic foot muscles would prevent injuries. It is a fallacy and must be amended.

3. Line 39: Please consider including other papers that presented different foot muscle exercises program, as Matias et al. BMC Musculoskelet Disord. 2016 doi: 10.1186/s12891-016-1016-9. and Ferreira et al. Trials. 2020 doi: 10.1186/s13063-019-4017-9.

4. Lines 43-44 : “These factors are believed to protect runners from common running-related injuries [21, 23]…” it is an incorrect affirmation. Reference 21 does not “believe”, it proved that foot-related exercises do prevent RRI and reference 23 did not study RRI. Consider rephrasing the sentence.

5. Lines 47-48: “… and transmission efficiency of GRF.” What did the authors mean by efficiency GRF?! It does not make sense the way it is written. Please, rephrase the sentence.

6. Lines 48-49: “Nonetheless, current studies on IFM training are cross-sectional, and evidence to support this claim is lacking.” This sentence is also a fallacy. There are many papers that as RCT and not cross-sectional and those RCTs have proven the efficacy of the foot muscles training for several conditions Just to name two: Mølgaard CM, J Sci Med Sport. 2018 doi: 10.1016/j.jsams.2017.05.019. And your references 10 and 21, among other that need to be reviewed.

7. It is not clear what is the aim of this systematic review: look for RCTs, since the authors criticized that the majority of current studies are cross-sectional, or focuses on all types of studies. This need to be clear in the introduction.

8. The main question investigated in the review must be clearly posted in the introduction and it is not clear at all.

Methods

1. The type of studies that were searched and included need to be described in the search strategy.

2. What is a “desired foot biomechanics outcome” itemized in (2) in the Study selection session.

3. Please, specify better what “outcome characteristics” are in the data extraction session.

4. There is a critical point in the methodology that is pooling data from studies that used different intervention approaches and time of intervention, and thus, the results are not expected to be the same. Pooling the data into a meta-analysis, it groups studies even if the results are not coming from the same approach of IFM training. This should be acknowledged and maybe exclude the meta-analysis of the manuscript. Or, at least, include as a limitation of the meta-analysis performed.

5. Figure 1 need to clearly describe the reasons for excluding the 204 papers from the review.

Results

1. “one article as clinical pearl…” What the authors mean by that?

2. When describing the meta-analysis, the muscle volume that changed in one study did not appear and should at least be mentioned, not only the CSA, but muscle volume. The way it looks, readers will not know IFM changes foot muscles volume.

3. Please, correct Weight to body mass, because you express in kg.

4. Table 3 – include descriptions of abbreviations in the caption: FPI, COM, INT and BMI.

Discussion

1. Overall, the discussion is straightforward and focused, but the English language that needs revision.

2. Lines 208-216: it is important to emphasizes that the studies did see differences after the intervention in CSA and volume of the foot muscles. But the heterogeneous and critical meta-analysis performed did not detect any difference probably due to pooling papers with completely different interventions, exercises and times of intervention and populations.

3. The same comment above can be applied to third, fourth, fifth paragraphs of the discussion.

6. PLOS authors have the option to publish the peer review history of their article (what does this mean?). If published, this will include your full peer review and any attached files.

Reviewer #1: No

Reviewer #2: No

---

## [Author Response · Author response to Decision Letter 0]

29 Oct 2021

Review Comments to the Author

Reviewer #1:

The authors present a systematic review study on the effectiveness of interventions that aim to improve the functioning of the Intrinsic Foot Muscles. This work has potential relevance in the prevention of foot and ankle injuries and provides a better understanding of the functional anatomy of the foot and the effects of training thereon. The manuscript overall is of sound quality, however it needs some strict improvements in the methods section in order to become eligible for publication.

A: Thank you for your positive comments. Your comments were highly insightful and help us improve the quality of the manuscript. We have modified the methods section.

MAJOR ISSUES

Aims. The aim of the current study is somewhat unclear: “Nonetheless, current studies on IFM training are cross-sectional, and evidence to support this claim is lacking. Therefore, the aim of this systematic review is to identify and determine the effect of IFM training on foot function and dynamic postural balance” (line 48-50).

This needs to be further clarified. If all past studies are cross sectional, what will be the added value of this review? In fact, later you find 13 RCTs that have been done. Why do these not feature in the introduction?

A: Thank you for your suggestion. This sentence needs further consideration and we have rewritten it as follows: 

“On the basis of previous findings, IFMs training can ameliorate foot biomechanic variables, resulting in the minimization of an accumulation of repetitive stresses and higher absorption of ground reaction force. Although numerous studies explore the effect of IFMs training, there is still lacking of evidence-based studies to review it systematically. Therefore, this systemic review with meta-analysis aimed to identify and determine the IFMs training on foot function and dynamic postural balance.” (Page 3 Line 51-55)

This issue is further confused at the start of the discussion where the aim seems to have been to summarize the current studies. Furthermore, judging from the methods and the results, the aim seems to have been to assess the effect of IFM training on foot biomechanical outcomes using a meta-analysis. Please clarify.

A: Thank you for your suggestion. We have modified it as follows:

“This systematic review summarized the current studies that explored the effect of IFMs training on foot biomechanical outcomes using a meta-analysis.” (Page 16 Line 229-230)

Methods. The major issues of the current manuscript lie in the methods section. Although the authors provide an adequate description of the methods to see that their approach is well thought out, there is insufficient details, and sometimes inconsistencies, that limit the reproducibility of the approach. Relating to this, the following issues will need to be addressed:

1. In the description of the search (line 57-60), it would be good to add some specification of the BOOLEAN operators used in the search.

A: We have added specification of the BOOLEAN operators in the manuscript. (Page 3 Line 62-67)

2. In the supplementary file specifying the search, it should be specified what type of search was performed (e.g. title/abstract/keywords, topic, mesh terms) and it should be specified how the search was adapted to the different search engines.

A: Thank you for your suggestion. We have revised it and given an example of search strategy for PubMed database.

3. There is some confusion in the use of quotation marks. In the main text, single quotation marks are used, in the supplementary file these are double marks. Please specify what was used in the searches, as these have a different meaning in the search engines.

A: Thank you for your suggestion. We have corrected the quotation marks. (Page 3 Line 62-67)

4. The authors note the inclusion criterium ‘Research specific to IFM training as an intervention’ (line 66). As the authors mention, these muscles have a function many whole-body activities (e.g. walking), so arguably can be trained with many whole body activities as well (e.g. walking, running, whole body vibration training). How was IFM training defined for inclusion in the review?

A: Thank you for your suggestion. IFMs training was defined as “training program emphasizing the neuromuscular recruitment of the plantar intrinsic foot muscles, like short foot exercise.” We have added it in the manuscript. (Page 4 Line 71-73)

5. No limitations were introduced to the search dates and yet only articles from after 2013 were included. This might just result from the novelty within the research field, but it could also indicate a fatal flaw in the search strategy. That is, it might be that the field evolved and that now different terms are used compared to 20 years ago. On example might be the following study: DOI: 10.1123/jsr.21.4.327. Please comment on this issue.

A: Thank you for your suggestion. In order to keep the novelty within the research field, we only included studies from 1 January 2011 to 14 February 2021. We have updated the search dates. (Page 3 Line 60-61)

For DOI: 10.1123/jsr.21.4.327, the main aim of this study was comparing the short-foot exercise and towel-curl exercise, rather than explore the effect of IFMs training on foot function or dynamic postural balance, so we excluded it.

6. The authors state ‘When possible, the results of the included studies were pooled for meta-analysis using RevMan 5.3.’ (line 84). How was it determined whether this was possible? It would be helpful to inform the reader on this process. How come only 6 of the 15 papers ended up in the meta-analysis?

A: Thank you for your suggestion. Because of the foot biomechanics are various, after reviewing all included articles, only some common parameters can be pooled. 

We have modified the sentence to “The pooled mean difference of training effects was calculated and illustrated by forest plots with Review Manager version 5.3. Random effects models were used to calculate standardized mean differences and 95% confidence intervals (CIs) for the control and experimental groups.” (Page 5 Line 92-94)

7. The prospero registration introduces the usage of the PEDro scale, in the methods section a ‘Checklist for Measuring Quality’ is introduced and in the results the Modified Downs and Black Quality Index is reported. Please resolve these inconsistencies and please specify why the prospero protocol was not adhered to.

A: Thank you for your suggestion. ‘Checklist for Measuring Quality’ is refers to the ‘Modified Downs and Black Quality Index’. We have corrected it. Additionally, the study was registered in PROSPERO (CRD42021232984), we added it in the supporting information. (Page 4 Line 84)

MINOR ISSUES

1.The use of abbreviations drastically limits the readability of the manuscript. Following PLOS one author guidelines, the use of abbreviations should be limited, and non-standard abbreviations should not be used unless they appear at least three times in the text. While I believe that most abbreviations are used more than thrice, this is mainly through their use in table 2. I would advise to limit abbreviating terms in the main text and only use abbreviations in the table.

A: Thank you for your suggestion. We have modified the abbreviated terms in the main text.

2. Line 22. I would suggest to use the term ‘effector’ rather than ‘organ’

A: We have revised " organ " as " effector ". 

3. Line 27. With the introduction of IFMs, I would advise to name which muscles are meant specifically.

A: We have named the potential muscles specifically. (Page 2 Line 29-30)

4. Line 39-40. With introduction of the training methods, it would be helpful if the authors could explain in a bit more detail what these training methods are about and in particular where they are different from each other.

A: Thank you for your suggestion. We have added details about the training methods as follows:

“Several methods can be used to train IFMs, such as short foot exercise (SFE), toe-posture exercises, towel curl exercises and metatarsophalangeal joint (MPJ) muscle training [17-21], among which, SFE was studied most because of it characteristics utilizing the IFMs to draw the metatarsal heads back towards the heel whilst minimizing distal interphalangeal flexion” (Page 2 Line 43-46)

5. Line 65-66. “The remaining studies were then reviewed as full texts on the basis of the following criteria: …” - The present wording makes it appear like the title and abstract screening were based on different criteria. Please clarify whether this was the case.

A: Thank you for your suggestion. Title, abstract and full texts screening all based on the same criteria. 

We have revised the sentence to “After removing duplicate articles, the search results were screened independently by two authors according to titles, abstracts and full texts on the basis of the following criteria.” (Page 3 Line 70-71)

6. Line 91-92. “After full text screening, one article as clinical pearl was excluded” – personally I am unfamiliar with the term ‘clinical pearl’ and why this should lead to exclusion. This might be just me, but in any case a bit further clarification might be helpful.

A: Thank you for your suggestion. The relative references have been added in the text. 

REF: Vincent KR, Vincent HK. Use of Foot Doming for Increasing Dynamic Stability and Injury Prevention in Runners and Athletes. Curr Sports Med Rep, 2018;17(10):320-321.

7. Line 135. Descriptive stats are only provided for runners. How about the non-runner participants?

A: Thank you for your suggestion. After re-reviewed the included studies, we added descriptive data of participants as follows:

“The 15 studies included a total of 610 participants. Six studies included elite long-distance runners (n=328) [17, 18, 22, 23, 37, 38]. Three studies explored IFMs on pes planus patients (n=75) [11, 35, 39], and six other studies only included healthy or asymptomatic subjects (n=207) [10, 16, 19, 25, 27, 36].” (Page 8 Line 152-154)

8. Line 148-149. The information about intervention duration is duplicated after section 3.2 and does not need to be mentioned here.

A: We have removed it.

9. Line 153. Please define extrinsic foot muscles.

A: Thank you for your suggestion. The extrinsic foot muscles are around the lower leg and act to plantarflexion dorsiflexion, invert and evert the foot. We have modified it. (Page 9 Line 171)

10. Line 169. Please define CSA

A: We have revised CSA as cross-sectional area

11. Line 199. Please provide a more extensive/descriptive caption for figure 4.

A: We have provided descriptive caption for figure 4.

Figure 4. Funnel plots showing publication bias among studies used to compare IFMs training and control groups (Page 11 Line 226-227)

12. Line 216. The authors mention advance technologies. Can they provide some example of such technologies?

A: We have provided example of advance technologies as follows: 

“future studies should utilise more advanced technology, like magnetic resonance imaging to measure the IFMs fat infiltration and cross-sectional area after training.” (Page 17 Line 242-243)

13. Line 238. I believe this should be ‘though’ instead of ‘through’.

A: We have revised " through’" as " though". 

Reviewer #2: 

The goal of this manuscript was to develop a systematic review and a meta-analysis systematic review to identify the effect of intrinsic foot muscles training on foot function and dynamic postural balance.

The manuscript is relatively well grounded, it deals with a relevant topic to be investigated and deserves to be published. In its current state, the level of English throughout your manuscript need careful revision. I suggest a fluent English speaker read and correct grammatical, word choice, and syntax errors throughout the manuscript.

Although paper is straightforward but there are many suggestions/comments that I addressed throughout the manuscript.

A: Thank you for your positive comments. Your comments were highly insightful and help us improve the quality of the manuscript. The level of English throughout our manuscript have been carefully revised. A native speaker was invited to edit the manuscript.

There is a critical point in the methodology that is pooling data from studies that used different intervention approaches and time of intervention, and thus, the results are not expected to be the same. Pooling the data into a meta-analysis, it groups studies even if the results are not coming from the same approach of IFM training. Pooling data that came from different interventions, exercises and times of intervention and populations lead to huge heterogeneity that will certainly compromise a meta-analysis. It is pivotal to choose same interventions to be included in a meta-analysis and only then we could have an unbiased statistical analysis. This should be acknowledged and maybe exclude the meta-analysis from the manuscript. it does not add much as the only two variables that could be pooled did not achieve power enough to conclude anything robust.

A: Thank you for your suggestion. Given that pooling the data that came from different methods may lead to huge heterogeneity and compromise a meta-analysis, after review the included studies again, only two variables that could be pooled in the current studies.

Language issues examples (there are other weird sentences and word choices that need revision):

1. Please, consider reviewing the English language and grammar by a native reviewer. The writing compromises the understanding of parts of the manuscript.

A: Thank you for your suggestion. The manuscript language and grammar have been carefully revised by a native reviewer. 

2. Introduction: “Whilst running, the feet act as the starting body parts…” – correct to body PART.

A: We have corrected it. (Page 2 Line 24)

3. Introduction: “Given that the feet are the most proximal aspect of the lower limb…”- correct to most DISTAL. The feet are distal and not proximal part of LL.

A: Thank you for your suggestion. We have corrected it. (Page 2 Line 27)

4. Introduction: “ground impact force (GRF)…” – correct to ground REACTION force.

A: We have corrected it. (Page 2 Line 28)

5. Introduction: “During the early stance phase in running…” – correct to OF REARFOOT STRIKE running

A: We have corrected it. (Page 2 Line 36)

6. Introduction: “… and providing better propulsion for runners in the swing phase [12].” – it provides better propulsion in the PUSH OFF phase, not swing phase!

A: Thank you for your suggestion. This is a typo. We have corrected it. (Page 2 Line 40)

Abstract and Introduction

1. Please, provide the PROSPERO registry number in the abstract.

A: Thank you for your suggestion. We have added registry number in the abstract. (Page 1 Line 15)

2. At the end of the second paragraph of the introduction, the authors provide a closing sentence concluding all previous sentences, however it does not make sense. “Therefore, strengthening of IFMs has been suggested for the prevention and treatment of foot and ankle sports injuries [16].” The previous sentences describe all important roles of the foot and its intrinsic muscles, but they do not provide any evidence that strengthening the intrinsic foot muscles would prevent injuries. It is a fallacy and must be amended.

A: Thank you for your suggestion. We have amended it.

3. Line 39: Please consider including other papers that presented different foot muscle exercises program, as Matias et al. BMC Musculoskelet Disord. 2016 doi: 10.1186/s12891-016-1016-9. and Ferreira et al. Trials. 2020 doi: 10.1186/s13063-019-4017-9.

A: Thank you for your suggestion. We have added the relative references in the text.

4. Lines 43-44: “These factors are believed to protect runners from common running-related injuries [21, 23]” it is an incorrect affirmation. Reference 21 does not “believe”, it proved that foot-related exercises do prevent RRI and reference 23 did not study RRI. Consider rephrasing the sentence.

A: Thank you for your suggestion. We have rephrased this sentence as follows: 

“which proved that foot-related exercises do prevent running-related injuries [23]” (Page 3 Line 48)

5. Lines 47-48: “… and transmission efficiency of GRF.” What did the authors mean by efficiency GRF?! It does not make sense the way it is written. Please, rephrase the sentence.

A: Thank you for your suggestion. We have rephrased this sentence as follows:

“resulting in the minimization of an accumulation of repetitive stresses and higher absorption of ground reaction force.” (Page 3 Line 52-53)

6. Lines 48-49: “Nonetheless, current studies on IFM training are cross-sectional, and evidence to support this claim is lacking.” This sentence is also a fallacy. There are many papers that as RCT and not cross-sectional and those RCTs have proven the efficacy of the foot muscles training for several conditions Just to name two: Mølgaard CM, J Sci Med Sport. 2018 doi: 10.1016/j.jsams.2017.05.019. And your references 10 and 21, among other that need to be reviewed.

A: Thank you for your suggestion. This sentence needs further consideration and we have rewritten it as follows:

“On the basis of previous findings, IFMs training can ameliorate foot biomechanic variables, resulting in the minimization of an accumulation of repetitive stresses and higher absorption of ground reaction force. Although numerous studies explore the effect of IFMs training, there is still lacking of evidence-based studies to review it systematically. Therefore, this systemic review with meta-analysis aimed to identify and determine the IFMs training on foot function and dynamic postural balance.” (Page 3 Line 52-55)

7. It is not clear what is the aim of this systematic review: look for RCTs, since the authors criticized that the majority of current studies are cross-sectional, or focuses on all types of studies. This need to be clear in the introduction.

A: Thank you for your suggestion. Some sentence needs further consideration in the introduction and we have rephrased this part. (Page 3 Line 52-55)

8. The main question investigated in the review must be clearly posted in the introduction and it is not clear at all.

A: Thank you for your suggestion. We have rewritten the review question clearly. 

Methods

1.The type of studies that were searched and included need to be described in the search strategy.

A: Thank you for your suggestion. The type of studies has been added in the search strategy. (Page 3 Line 66)

2. What is a “desired foot biomechanics outcome” itemized in (2) in the Study selection session.

A: Thank you for your suggestion. “biomechanics outcome” means common biomechanics parameters in the studies. We have corrected it. (Page 4 Line 73)

3. Please, specify better what “outcome characteristics” are in the data extraction session.

A: Thank you for your suggestion. In the “outcome characteristics”, we have modified it. (Page 4 Line 79)

4. There is a critical point in the methodology that is pooling data from studies that used different intervention approaches and time of intervention, and thus, the results are not expected to be the same. Pooling the data into a meta-analysis, it groups studies even if the results are not coming from the same approach of IFM training. This should be acknowledged and maybe exclude the meta-analysis of the manuscript. Or, at least, include as a limitation of the meta-analysis performed.

A: Thank you for your suggestion. We admitted that different intervention approaches and time of intervention may contribute to the results different from what we expected. We have modified it in the limitation. (Page 19 Line 281-282)

5. Figure 1 need to clearly describe the reasons for excluding the 204 papers from the review.

A: We have described the reasons for excluding papers clearly. (Page 6 Line 116-119)

Results

1.“one article as clinical pearl…” What the authors mean by that?

A: Thank you for your suggestion. The relative references have been added in the text. 

REF: Vincent KR, Vincent HK. Use of Foot Doming for Increasing Dynamic Stability and Injury Prevention in Runners and Athletes. Curr Sports Med Rep, 2018;17(10):320-321. (Page 5 Line 100-101)

2.When describing the meta-analysis, the muscle volume that changed in one study did not appear and should at least be mentioned, not only the CSA, but muscle volume. The way it looks, readers will not know IFM changes foot muscles volume.

A: Thank you for your suggestion. We have added muscle volume change in the manuscript. (Page 9 Line 187)

3. Please, correct Weight to body mass, because you express in kg.

A: We have corrected descriptions Weight to body mass. 

4. Table 3 – include descriptions of abbreviations in the caption: FPI, COM, INT and BMI.

A: Thank you for your suggestion. We have included descriptions of abbreviations. 

Discussion

1.Overall, the discussion is straightforward and focused, but the English language that needs revision.

A: Thank you for your suggestion. We have asked for native English speaker to revise our manuscript. 

2.Lines 208-216: it is important to emphasizes that the studies did see differences after the intervention in CSA and volume of the foot muscles. But the heterogeneous and critical meta-analysis performed did not detect any difference probably due to pooling papers with completely different interventions, exercises and times of intervention and populations.

A: Thank you for your suggestion. Some studies may be see differences after the intervention in CSA and volume of the foot muscles. But the heterogeneous and critical meta-analysis performed actually did not detect any difference.

3. The same comment above can be applied to third, fourth, fifth paragraphs of the discussion.

A: Thank you for your suggestion. IFMs training may bring some positive effect of foot biomechanics parameters, but after the pooled data analysis of the included studies, the actual effect of strength training on IFMs was various.

---

## [Decision Letter · Decision Letter 1]

9 Dec 2021

PONE-D-21-23366R1Effect of intrinsic foot muscles training on foot function and dynamic postural balance: A systematic review and meta-analysisPLOS ONE

Dear Dr. Wang,

Thank you for submitting your manuscript to PLOS ONE. After careful consideration, we feel that it has merit but does not fully meet PLOS ONE’s publication criteria as it currently stands. Therefore, we invite you to submit a revised version of the manuscript that addresses the points raised during the review process. The language has to be cleaned up, before the paper can be further considered for publication.

We look forward to receiving your revised manuscript.

Kind regards,

Peter Andreas Federolf

Academic Editor

PLOS ONE

Reviewers' comments:

Reviewer's Responses to Questions

**Comments to the Author**

1. If the authors have adequately addressed your comments raised in a previous round of review and you feel that this manuscript is now acceptable for publication, you may indicate that here to bypass the “Comments to the Author” section, enter your conflict of interest statement in the “Confidential to Editor” section, and submit your "Accept" recommendation.

Reviewer #2: (No Response)

2. Is the manuscript technically sound, and do the data support the conclusions?

Reviewer #2: Yes

3. Has the statistical analysis been performed appropriately and rigorously? 

Reviewer #2: Yes

4. Have the authors made all data underlying the findings in their manuscript fully available?

Reviewer #2: Yes

5. Is the manuscript presented in an intelligible fashion and written in standard English?

Reviewer #2: No

6. Review Comments to the Author

Reviewer #2: 1. Although most of the previous comments from this reviewer were addressed properly, the paper still needs improvement in language and flow. It is impossible to follow some new additions. I strongly recommend a professional service for proofreading, not only a colleague who is a native speaker, because it clearly did not work this time. The way the paper reads it is impossible to be fully understood.

2. I have used the search strategies now described in the paper and found some very important quality papers (PEDro scale 5 to 8) that were not included in this review. Additionally, I am familiar with many papers that used IFM training and it matched some of them that I found but is it no in this manuscript. I am listing below for the authors to consider including as it must not be out of your systematic review.

Missing relevant papers regarding you research question, your outcomes and your chosen study designs:

I. (PEDro score 7) Lee D-R, Choi Y-E. Effects of a 6-week intrinsic foot muscle exercise program on the functions of intrinsic foot muscle and dynamic balance in patients with chronic ankle instability. Journal of exercise rehabilitation. 2019;15(5):709.

Specifically, you cite this study but I did not understand why it was not included in your systematic review.

II. (PEDro score 5) Day EM, Hahn ME. Increased toe-flexor muscle strength does not alter metatarsophalangeal and ankle joint mechanics or running economy. Journal of sports sciences. 2019;37(23):2702-10.

III. (PEDro score 8) Jung D-Y, Koh E-K, Kwon O-Y. Effect of foot orthoses and short-foot exercise on the cross-sectional area of the abductor hallucis muscle in subjects with pes planus: a randomized controlled trial. Journal of back and musculoskeletal rehabilitation. 2011;24(4):225-31.

IV. (PEDro score 8) Kamonseki DH, Gonçalves GA, Liu CY, Júnior IL. Effect of stretching with and without muscle strengthening exercises for the foot and hip in patients with plantar fasciitis: a randomized controlled single-blind clinical trial. Manual therapy. 2016;23:76-82. (balance)

V. (PEDro score 6) Lynn SK, Padilla RA, Tsang KK. Differences in static-and dynamic-balance task performance after 4 weeks of intrinsic-foot-muscle training: the short-foot exercise versus the towel-curl exercise. Journal of sport rehabilitation. 2012;21(4):327-33.

3. There are two studies included that have a wrong year of publication. Please, correct it:

• Unver B, Erdem EU, Akbas E. Effects of Short-Foot Exercises on Fo 506 ot Posture, Pain, Disability and Plantar Pressure in Pes Planus. Journal of sport rehabilitation. 2019:1-16.

• Fraser JJ, Hertel J. Effects of a 4-week intrinsic foot muscle exercise program on motor function: a preliminary randomized control trial. Journal of sport rehabilitation. 2019;28(4):339-49.

4. In the PROSPERO registry, the risk of bias assessment is described as PEDro scale. However, in this manuscript the authors did no use this scale but the Modified Downs and Black Quality Index. Please resolve this inconsistency with your PROSPERO registry and specify why the PROSPERO protocol was not adhered to. The reviewer #1 had already pointed it out in the first review but it seems the authors did not understand it and have not explained this issue.

5. Ref 35 is listed as “Kim EK, Kim JS. 2016” As authors but in the evidence table, it is listed as “Eun-Kyung et al. (2016)” Please correct that.

6. Introduction Lines 53-54: ““Although numerous studies explore the effect of IFMs training, there is still lacking of evidence-based studies to review it systematically.” This sentence does not make sense. If there are many evidences, and it does indeed (you mentioned now in your paper) that proved the efficacy of the foot muscles training for several conditions, what exactly do you need? There is no “lacking evidenced-based studies”. On the contrary. What you really want is to review systematically the exiting evidences. Please, rephrase, again, the sentence.

7. Introduction Lines 66-67: “Randomized controlled trials, intervention studies, were eligible for inclusion.” What are intervention studies? Non-controlled studies? Case studies? Interrupted case series? Please, use the SIGN (Scottish Intercollegiate Guidelines Network) methodology to describe the type of studies that were searched and included in your review

7. PLOS authors have the option to publish the peer review history of their article (what does this mean?). If published, this will include your full peer review and any attached files.

Reviewer #2: **Yes: **Isabel C. N. Sacco

---

## [Author Response · Author response to Decision Letter 1]

11 Jan 2022

Reviewer #2: 

1. Although most of the previous comments from this reviewer were addressed properly, the paper still needs improvement in language and flow. It is impossible to follow some new additions. I strongly recommend a professional service for proofreading, not only a colleague who is a native speaker, because it clearly did not work this time. The way the paper reads it is impossible to be fully understood.

A: Thank you for your suggestion. We have modified the English language of the manuscript.

2. I have used the search strategies now described in the paper and found some very important quality papers (PEDro scale 5 to 8) that were not included in this review. Additionally, I am familiar with many papers that used IFM training and it matched some of them that I found but is it no in this manuscript. I am listing below for the authors to consider including as it must not be out of your systematic review. 

Missing relevant papers regarding you research question, your outcomes and your chosen study designs:

I. (PEDro score 7) Lee D-R, Choi Y-E. Effects of a 6-week intrinsic foot muscle exercise program on the functions of intrinsic foot muscle and dynamic balance in patients with chronic ankle instability. Journal of exercise rehabilitation. 2019;15(5):709.

Specifically, you cite this study but I did not understand why it was not included in your systematic review.

A: Thank you for your suggestion. Given to the fact that patients with chronic ankle instability may have different foot biomechanics, and IFMs training has less effect on the ankle joint. So after negotiation between the two authors, we finally decided excluded this article.

II. (PEDro score 5) Day EM, Hahn ME. Increased toe-flexor muscle strength does not alter metatarsophalangeal and ankle joint mechanics or running economy. Journal of sports sciences. 2019;37(23):2702-10. 

A: Thank you for your suggestion. After reexamined all articles, we found this article meets our inclusion criteria. We have added it and rewrite our manuscript.

III. (PEDro score 8) Jung D-Y, Koh E-K, Kwon O-Y. Effect of foot orthoses and short-foot exercise on the cross-sectional area of the abductor hallucis muscle in subjects with pes planus: a randomized controlled trial. Journal of back and musculoskeletal rehabilitation. 2011;24(4):225-31. 

A: Thank you for your suggestion. Because of this article mainly compare foot orthosis (FO) group and the group combined foot orthosis and short-foot exercise (FOSF) after 8-week intervention. We cannot rule out the effect of foot orthosis on the cross-sectional area of the abductor hallucis muscle. So after reviewing the titles, abstracts and full texts, we excluded it.

IV. (PEDro score 8) Kamonseki DH, Gonçalves GA, Liu CY, Júnior IL. Effect of stretching with and without muscle strengthening exercises for the foot and hip in patients with plantar fasciitis: a randomized controlled single-blind clinical trial. Manual therapy. 2016;23:76-82. (balance) 

A: Thank you for your suggestion. We only include research specific to intrinsic foot muscles training. The foot exercise group in this article utilizing extrinsic and intrinsic foot muscles. Given that the extrinsic foot muscles strengthening exercises may bring potential effect on the foot biomechanics outcome, so after reviewing the titles and abstracts, we excluded it.

V. (PEDro score 6) Lynn SK, Padilla RA, Tsang KK. Differences in static-and dynamic-balance task performance after 4 weeks of intrinsic-foot-muscle training: the short-foot exercise versus the towel-curl exercise. Journal of sport rehabilitation. 2012;21(4):327-33.

A: Thank you for your suggestion. The main aim of this study was to compare short-foot exercise and towel-curl exercise on foot function or dynamic postural balance, rather than explore the effect of IFMs training foot function and dynamic postural balance. So after reviewing the titles, abstracts and full texts, we excluded it.

3. There are two studies included that have a wrong year of publication. Please, correct it: 

• Unver B, Erdem EU, Akbas E. Effects of Short-Foot Exercises on Fo 506 ot Posture, Pain, Disability and Plantar Pressure in Pes Planus. Journal of sport rehabilitation. 2019:1-16. 

• Fraser JJ, Hertel J. Effects of a 4-week intrinsic foot muscle exercise program on motor function: a preliminary randomized control trial. Journal of sport rehabilitation. 2019;28(4):339-49.

A: Thank you for your suggestion. We have corrected it.

4. In the PROSPERO registry, the risk of bias assessment is described as PEDro scale. However, in this manuscript the authors did no use this scale but the Modified Downs and Black Quality Index. Please resolve this inconsistency with your PROSPERO registry and specify why the PROSPERO protocol was not adhered to. The reviewer #1 had already pointed it out in the first review but it seems the authors did not understand it and have not explained this issue. 

A: Thank you for your suggestion. Even though we plan to utilize PEDro scale to assess the risk of bias, we finally decide to use Modified Downs and Black Quality Index to assess the methodological quality of the included studies based on the relative references, and we have added it to support our manuscript.

Ref:

1. Huffer D, Hing W, Newton R, Clair M. Strength training for plantar fasciitis and the intrinsic foot musculature: A systematic review. Phys Ther Sport, 2017; 24:44-52. 

2. Downs SH, Black N. The feasibility of creating a checklist for the assessment of the methodological quality both of randomised and non-randomised studies of health care interventions. J Epidemiol Commun H, 1998. 

5. Ref 35 is listed as “Kim EK, Kim JS. 2016” As authors but in the evidence table, it is listed as “Eun-Kyung et al. (2016)” Please correct that.

A: Thank you for your suggestion. We have corrected it.

6. Introduction Lines 53-54: ““Although numerous studies explore the effect of IFMs training, there is still lacking of evidence-based studies to review it systematically.” This sentence does not make sense. If there are many evidences, and it does indeed (you mentioned now in your paper) that proved the efficacy of the foot muscles training for several conditions, what exactly do you need? There is no “lacking evidenced-based studies”. On the contrary. What you really want is to review systematically the exiting evidences. Please, rephrase, again, the sentence. 

A: Thank you for your suggestion. We have rephrased this sentence.

7. Introduction Lines 66-67: “Randomized controlled trials, intervention studies, were eligible for inclusion.” What are intervention studies? Non-controlled studies? Case studies? Interrupted case series? Please, use the SIGN (Scottish Intercollegiate Guidelines Network) methodology to describe the type of studies that were searched and included in your review.

A: Thank you for your suggestion. We have modified the description the type of studies according to the SIGN (Scottish Intercollegiate Guidelines Network) methodology, and added the relative reference to support our review.

Ref:

Scottish Intercollegiate Guidelines Network (SIGN) Methodology Review Group, editor. Report on the review of the method of grading guideline recommendations. Edinburgh: SIGN 1999.

---

## [Decision Letter · Decision Letter 2]

10 Feb 2022

PONE-D-21-23366R2Effect of intrinsic foot muscles training on foot function and dynamic postural balance: A systematic review and meta-analysisPLOS ONE

Dear Dr. Wang,

Your paper has been assessed by 2 reviewers of whom one is satisfied with the changes. The other reviewer still has several major and minor comments. 

I will allow one more revision. Please address the remaining issues carefully.

We look forward to receiving your revised manuscript.

Kind regards,

Peter Andreas Federolf

Academic Editor

PLOS ONE

Additional Editor Comments:

Comments of Reviewer 2:

The authors have written an interesting review paper that, through the peer-review process so far has made some obvious progress. Still, some major issues remain, as well as numerous minor issues. Major issues lie in the reproducibility and complexity of the methods and results. Still significant steps are required in these sections before they would be fit for publication. Further, there needs to be a clear match between the results collected here and the conclusions of the study, as the discussion section currently reads like a narrative literature review, without adding much analyses to the reviewed studies (the analyses seem isolated in the results and are not clearly linked to the discussion).

I appreciate the further specification of the authors decision for using the Modified Downs and Black quality assessment to the manuscript. However, I do not feel the authors have sufficiently explained why they have neglected an a-priori established part of their protocol and chosen to use a different method. This undermines the credibility of the manuscript as one of the major strengths of a systematic review lies in the protocol that is established a-priori. As such, I would again ask the authors to explain in their response to reviewers letter why they made this decision.

Further comments in order of appearance.

Line 3-5. This sentence does not flow well due to the double use of the word ‘relevant’. I would suggest the following sentence: ‘Keywords related to IFM training were used to search four databases for relevant studies published between January 2011 and February 2021.’

Line 34-36. This sentence does not flow well as it is unclear how ‘they’ should be interpreted. I would suggest rephrasing to something starting with: ‘The main IFMs are … ‘

Line 45. So far this paragraph has been about running, but this final sentence switches the focus to walking. This breaks up the flow in the introduction.

Line 49-52. There is a flaw in the logic here. It cannot be stated that the part up until the first reference ‘[24]’ is proof for the part until the second reference ‘[23]’. These findings might be related, but proof is not offered by this statement.

Line 55-56. The aims statement does not flow logically from the previous. First, you state that much about the positive effects of IFM is already known. This is not a reason to study the concept further. A better rationale could be something along the lines of the following: ‘while a number of isolated studies have shown benefits of IFM, the applicability of these findings is still limited because, to date, no previous study has systematically studied these effects nor has a meta-analysis been applied to get an overall estimate of the effect of IFM training. Therefore, the current study aims…’

Line 72. Consider changing ‘in accordance with’ to ‘based on’

Line 74-75. Please specify here what ‘desired foot biomechanical parameters’ were

Line 80. Consider ‘i.e.’ to ‘e.g.’

Line 87-91. This statement is confusing and needs to be rephrased.

Line 92-97. It is clear that the mean difference was taken as an outcome measure. However, it is unclear what this difference is based on, related to the different designs included in the study. For instance with an RCT. Was the difference of the experimental group between pre and post test used, or the difference between experimental and control group at post test?

Line 170. Please report exact p values and associated tests statistics.

Line 170-172. This sentence reads like a concluding sentence and therefore does not fit the results section.

Line 178-180. The authors conclude here that the funnel plot shows a ‘slight asymmetry’. What was the basis for this classification? That is, as a rule of thumb, 10 observations are required to statistically assess the asymmetry of a funnel plot. Unless the authors are aware of other guidelines, I think caution is warranted in this interpretation step. Especially considering that if one would scale figure 4B to have a X axis from -6 to 6, the asymmetry would appear way more drastic.

Line 184. The Eun-Kyung (2016) reference is not represented in table 1 and 2.

Line 184-187. This figure needs a clearer caption. What do the values indicate? Especially the ‘total’ column is not clear. I initially interpreted this as the N of the studies, but if this is true, then there are inconsistencies with table 2.

Table 3. Please specify the meaning of ‘NR’

Line 225. Which hypothesis is meant here?

Discussion in general: this section reach much like a narrative review and the link to the quantitative results of this study should be emphasized.

Line 179. Consider using ‘supervisor’ rather than ‘supervisors’

Reviewers' comments:

Reviewer's Responses to Questions

**Comments to the Author**

1. If the authors have adequately addressed your comments raised in a previous round of review and you feel that this manuscript is now acceptable for publication, you may indicate that here to bypass the “Comments to the Author” section, enter your conflict of interest statement in the “Confidential to Editor” section, and submit your "Accept" recommendation.

Reviewer #1: (No Response)

Reviewer #2: All comments have been addressed

2. Is the manuscript technically sound, and do the data support the conclusions?

Reviewer #1: Partly

Reviewer #2: Yes

3. Has the statistical analysis been performed appropriately and rigorously? 

Reviewer #1: Yes

Reviewer #2: Yes

4. Have the authors made all data underlying the findings in their manuscript fully available?

Reviewer #1: Yes

Reviewer #2: Yes

5. Is the manuscript presented in an intelligible fashion and written in standard English?

Reviewer #1: No

Reviewer #2: Yes

6. Review Comments to the Author

Reviewer #1: (No Response)

Reviewer #2: The additions and corrections were properly executed and the paper sounds better in this version.

7. PLOS authors have the option to publish the peer review history of their article (what does this mean?). If published, this will include your full peer review and any attached files.

Reviewer #1: No

Reviewer #2: **Yes: **Isabel C N Sacco

---

## [Author Response · Author response to Decision Letter 2]

24 Feb 2022

Further comments in order of appearance. 

Line 13-15. This sentence does not flow well due to the double use of the word ‘relevant’. I would suggest the following sentence: ‘Keywords related to IFM training were used to search four databases for relevant studies published between January 2011 and February 2021.’ 

A: Thank you for your suggestion. We have revised our manuscript follow your suggestion. 

Line 34-36. This sentence does not flow well as it is unclear how ‘they’ should be interpreted. I would suggest rephrasing to something starting with: ‘The main IFMs are … ‘ 

A: Thank you for your suggestion. We have rephrased our manuscript and start with: ‘The main IFMs are …’

Line 45. So far this paragraph has been about running, but this final sentence switches the focus to walking. This breaks up the flow in the introduction. 

A: Thank you for your suggestion. We have revised it.

Line 49-52. There is a flaw in the logic here. It cannot be stated that the part up until the first reference ‘[24]’ is proof for the part until the second reference ‘[23]’. These findings might be related, but proof is not offered by this statement. 

A: Thank you for your suggestion. We have revised it.

Line 55-56. The aims statement does not flow logically from the previous. First, you state that much about the positive effects of IFM is already known. This is not a reason to study the concept further. A better rationale could be something along the lines of the following: ‘while a number of isolated studies have shown benefits of IFM, the applicability of these findings is still limited because, to date, no previous study has systematically studied these effects nor has a meta-analysis been applied to get an overall estimate of the effect of IFM training. Therefore, the current study aims…’ 

A: Thank you for your suggestion. We have revised our manuscript follow your suggestion.

Line 72. Consider changing ‘in accordance with’ to ‘based on’ 

A: We have corrected it.

Line 74-75. Please specify here what ‘desired foot biomechanical parameters’ were 

A: Thank you for your suggestion. We have specified it.

Line 80. Consider ‘i.e.’ to ‘e.g.’ 

A: We have corrected it.

Line 87-91. This statement is confusing and needs to be rephrased. 

A: Thank you for your suggestion. We have rewritten it.

Line 92-97. It is clear that the mean difference was taken as an outcome measure. However, it is unclear what this difference is based on, related to the different designs included in the study. For instance with an RCT. Was the difference of the experimental group between pre and post test used, or the difference between experimental and control group at post test? 

A: Thank you for your suggestion. The mean difference is based on the difference between pre and post test used. 

Line 170. Please report exact p values and associated tests statistics. 

A: Thank you for your suggestion. We have reported exact p values. 

Line 170-172. This sentence reads like a concluding sentence and therefore does not fit the results section. 

A: We have deleted it.

Line 178-180. The authors conclude here that the funnel plot shows a ‘slight asymmetry’. What was the basis for this classification? That is, as a rule of thumb, 10 observations are required to statistically assess the asymmetry of a funnel plot. Unless the authors are aware of other guidelines, I think caution is warranted in this interpretation step. Especially considering that if one would scale figure 4B to have a X axis from -6 to 6, the asymmetry would appear way more drastic. 

A: Thank you for your suggestion. We have modified our manuscript following your suggestion.

Line 184. The Eun-Kyung (2016) reference is not represented in table 1 and 2. 

A: Thank you for your suggestion. Eun-Kyung (2016) has been revised as Kim et al [37] in table 1-3.

Line 184-187. This figure needs a clearer caption. What do the values indicate? Especially the ‘total’ column is not clear. I initially interpreted this as the N of the studies, but if this is true, then there are inconsistencies with table 2. 

A: Thank you for your suggestion. We have modified the caption of figure, the ‘total’ column is the N of the studies, this typo has been corrected.

Table 3. Please specify the meaning of ‘NR’ 

A: We have added it.

Line 225. Which hypothesis is meant here? Discussion in general: this section reach much like a narrative review and the link to the quantitative results of this study should be emphasized. 

A: Thank you for your suggestion. We have modified this section.

Line 279. Consider using ‘supervisor’ rather than ‘supervisors’ 

A: We have corrected it.

---

## [Decision Letter · Decision Letter 3]

23 Mar 2022

Effect of intrinsic foot muscles training on foot function and dynamic postural balance: A systematic review and meta-analysis

PONE-D-21-23366R3

Dear Dr. Wang,

We’re pleased to inform you that your manuscript has been judged scientifically suitable for publication and will be formally accepted for publication once it meets all outstanding technical requirements.

Kind regards,

Peter Andreas Federolf

Academic Editor

PLOS ONE

Reviewers' comments:

Reviewer's Responses to Questions

**Comments to the Author**

1. If the authors have adequately addressed your comments raised in a previous round of review and you feel that this manuscript is now acceptable for publication, you may indicate that here to bypass the “Comments to the Author” section, enter your conflict of interest statement in the “Confidential to Editor” section, and submit your "Accept" recommendation.

Reviewer #1: All comments have been addressed

2. Is the manuscript technically sound, and do the data support the conclusions?

Reviewer #1: Yes

3. Has the statistical analysis been performed appropriately and rigorously? 

Reviewer #1: Yes

4. Have the authors made all data underlying the findings in their manuscript fully available?

Reviewer #1: Yes

5. Is the manuscript presented in an intelligible fashion and written in standard English?

Reviewer #1: Yes

6. Review Comments to the Author

Reviewer #1: All of my previous comments have been addressed. To add one final comment: I would suggest adjusting the x-axes in figure 4 to have the same values in panel a and b. This will improve the readability and the reader's capability to compare both images.

7. PLOS authors have the option to publish the peer review history of their article (what does this mean?). If published, this will include your full peer review and any attached files.

Reviewer #1: **Yes: **Steven van Andel

---

## [Editor Report · Acceptance letter]

7 Apr 2022

PONE-D-21-23366R3 

Effect of intrinsic foot muscles training on foot function and dynamic postural balance:
A systematic review and meta-analysis 

Dear Dr. Wang:

I'm pleased to inform you that your manuscript has been deemed suitable for publication in PLOS ONE. Congratulations! Your manuscript is now with our production department. 

Kind regards, 

on behalf of

Dr. Peter Andreas Federolf 

Academic Editor

PLOS ONE